# Taxonomic and Gene Category Analyses of Subgingival Plaques from a Group of Japanese Individuals with and without Periodontitis

**DOI:** 10.3390/ijms22105298

**Published:** 2021-05-18

**Authors:** Kazuki Izawa, Kazuko Okamoto-Shibayama, Daichi Kita, Sachiyo Tomita, Atsushi Saito, Takashi Ishida, Masahito Ohue, Yutaka Akiyama, Kazuyuki Ishihara

**Affiliations:** 1Department of Computer Science, School of Computing, Tokyo Institute of Technology, Meguro-ku, Tokyo 152-8550, Japan; izawa@bi.c.titech.ac.jp (K.I.); ishida@c.titech.ac.jp (T.I.); ohue@c.titech.ac.jp (M.O.); akiyama@c.titech.ac.jp (Y.A.); 2Department of Microbiology, Tokyo Dental College, Chiyoda-ku, Tokyo 101-0061, Japan; okamotok@tdc.ac.jp; 3Department of Periodontology, Tokyo Dental College, Chiyoda-ku, Tokyo 101-0061, Japan; kitadaichi@tdc.ac.jp (D.K.); tomitas@tdc.ac.jp (S.T.); atsaito@tdc.ac.jp (A.S.); 4Oral Health Science Center, Tokyo Dental College, Chiyoda-ku, Tokyo 101-0061, Japan

**Keywords:** periodontal disease, periodontitis, subgingival plaque biofilm, periodontal pathogens, metagenomics, human oral microbiome

## Abstract

Periodontitis is an inflammation of tooth-supporting tissues, which is caused by bacteria in the subgingival plaque (biofilm) and the host immune response. Traditionally, subgingival pathogens have been investigated using methods such as culturing, DNA probes, or PCR. The development of next-generation sequencing made it possible to investigate the whole microbiome in the subgingival plaque. Previous studies have implicated dysbiosis of the subgingival microbiome in the etiology of periodontitis. However, details are still lacking. In this study, we conducted a metagenomic analysis of subgingival plaque samples from a group of Japanese individuals with and without periodontitis. In the taxonomic composition analysis, genus *Bacteroides* and *Mycobacterium* demonstrated significantly different compositions between healthy sites and sites with periodontal pockets. The results from the relative abundance of functional gene categories, carbohydrate metabolism, glycan biosynthesis and metabolism, amino acid metabolism, replication and repair showed significant differences between healthy sites and sites with periodontal pockets. These results provide important insights into the shift in the taxonomic and functional gene category abundance caused by dysbiosis, which occurs during the progression of periodontal disease.

## 1. Introduction

Periodontitis is a highly prevalent inflammatory and infectious disease of the tooth-supporting tissues. It is caused by bacteria present in the subgingival plaque (biofilm) and the host immune responses [1]. Imbalance in inflammatory and immune responses leads to the destruction of periodontal tissues, including alveolar bone loss, ultimately causing loss of teeth. The percentage of adults with periodontitis is very high. Approximately 42% of dentate adults between 30 and 79 years in the USA suffer from periodontitis [2], and the global age-standardized prevalence of severe periodontitis is approximately 10% [3]. Cumulative evidence has indicated that periodontal disease can underlie or exacerbate systemic diseases such as atherosclerosis, diabetes, rheumatoid arthritis [4], and Alzheimer’s disease [5]. These reports indicated that the eradication of periodontitis could greatly contribute to public health. However, to date, the etiologic agents of periodontitis are yet to be fully identified.

The results of the traditional investigations of the pathogens in the subgingival plaque, using methods such as culturing [6], DNA probes [7], and polymerase chain reaction (PCR) [8], have implicated gram-negative anaerobes such as *Porphyromonas gingivalis* and *Tannerella forsythia,* in the pathogenesis and progression of periodontitis. However, these conventional methods cannot provide a full picture of periodontal pathogens in subgingival plaque samples. Therefore, further comprehensive analysis of the subgingival microbiome is necessary. In addition, the results of animal experiments have shown increases in the bacterial species and their numbers following the infection by *P. gingivalis*, and a synergistic increase in the virulence of mixed infections by *P. gingivalis* and *Streptococcus gordonii* [9,10]. Based on these reports, dysbiosis of the subgingival microbiome is thought to be a major cause of periodontitis [11]. A comprehensive investigation of the microbiome shifts is essential to understand the mechanism underlying dysbiosis. Recently, several approaches for investigating the microbiome have been developed. One of these is next-generation sequencing, such as 16S rRNA sequencing and metagenome analysis, and another is a culture-based method, such as culturomics, which involves isolation of bacteria by culturing under different conditions and identification by using matrix-assisted laser desorption/ionization-time of flight mass spectrometry [12]. Among these, methods using next-generation sequencing have made comprehensive analysis of the whole microbiome of multiple samples possible. Recently, analyses using 16S rRNA sequencing have clarified the relative abundance of pathogens at the sites of periodontitis (periodontal pockets) [13,14]. 

During dysbiosis, a number of polymicrobial synergies occur [15]. In the subgingival plaques, more than 700 bacterial taxa have been detected [16]. Multiple species and genes may be involved in these synergies. Endo et al. [17] reported that essential genes involved in butyric acid metabolism within the fatty acid biosynthesis pathway are complemented in *P. gingivalis*, *T. forsythia*, and *Treponema denticola*. The composition of the microbiomes of several body areas, including supragingival plaque, buccal mucosa and tongue dorsum, showed diversity among subjects at the phylum level; however, the functional genes in this metabolic pathway were similar among all subjects [18]. These reports indicate the necessity of functional gene analysis. In addition, differences in the virulence within each species have been reported. The virulence of *P. gingivalis* with type II fimbriae was reportedly higher than that of the same species that produce other types of fimbriae [19]. A variant of *Aggregatibacter actinomycetemcomitans,* another major pathogen implicated in periodontitis, with mutations in its leukotoxin promoter region, produces significantly higher levels of leukotoxin than the original strain does [20]. These reports indicate that metagenomic analyses of subgingival plaque are essential to clarify the mechanisms of interspecies synergy inducing dysbiosis. Here, we present a metagenomic analysis of subgingival plaque samples from a group of Japanese individuals with or without periodontitis, aiming to clarify the taxonomic and functional genomic abundance in periodontitis.

## 2. Results 

### 2.1. Sequence Overview of Plaque Samples

We obtained 64 subgingival plaque samples from 33 participants (age range 25–79 years old) (Appendix A). After quality control of the reads, removal of the human genome, and extraction of the samples satisfying the threshold of the read number for metagenomic analysis, 42 samples were subjected to further data analysis (Table 1.).

### 2.2. Taxonomic Composition Analysis

Taxonomic compositions were estimated from fragments of 16S rDNA in metagenomic reads. Less than 1% relative abundance in all samples was categorized as “others”, and we identified 60 genera (Appendix A). We have summarized the top 25 genera for average of all samples in Figure 1. 

Taxonomic compositions varied among individuals and the disease status of the sites. At the healthy sites, *Corynebacterium, Actinomyces, Capnocytophaga*, *Fusobacterium, Rothia* and *Neisseria* were predominant (Appendix A). At the sites with periodontal pockets, *Fusobacterium, Porphyromonas, Prevotella Actinomyces, Treponema, Corynebacterium*, and *Neisseria* were predominant. *Bacteroides, Porphyromonas, Tannerella, Burkholderia, Moraxella, Treponema, Fretibacterium, Mogibacterium, Shuttleworthia, Filifactor*, and *Megasphaera* were increased with >3-fold higher abundance in the sites with periodontal pockets than the healthy sites (Appendix A, Appendix A). In addition, Genera including *Abiotrophia, Dialister, Filifactor* and *Fretibacterium* were increased with >3-fold abundance in the samples from healthy sites in periodontitis patient compared with that from the healthy sites in the healthy individuals (Appendix A). We further compared the abundances of 60 genera between healthy sites and sites with periodontal pockets using Mann-Whitney *U* test. The results from the taxonomic comparison indicated the change of abundance in the following two genera: *Bacteroides* (*p* = 0.047, power = 0.359) and *Mycobacterium* (*p* = 0.021, power = 0.397). The comparison of the mean abundance of these genera is shown in Figure 2. 

### 2.3. Functional Gene Category Analysis

The relative abundances of functional gene categories were estimated from metagenomic reads using the KEGG database (Figure 3). Compared to the taxonomic composition, the relative abundances of functional gene categories seemed to be uniform between individuals, especially at healthy sites. For a more detailed exploration, we again performed the Mann–Whitney *U* test for each of the 19 functional gene categories in healthy or periodontitis sampling sites. The results from this analysis indicated differences in the abundance of genes functioning in the following five categories: carbohydrate metabolism (*p* = 0.032, power = 0.328), amino acid metabolism (*p* = 0.035, power = 0.324), glycan biosynthesis and metabolism (*p* = 0.023, power = 0.347), and replication and repair (*p* = 0.035, power = 0.324). Comparisons of the mean abundances of these gene categories in healthy and periodontitis sites are shown in Figure 4.

Currently, little is known about the decrease in amino acid metabolism, and the increase in the replication and repair in periodontitis sites. For a more detailed exploration of these three categories, we conducted the Mann-Whitney *U* test (*p* < 0.05) on one lower-level category. From lower levels of amino acid metabolism, a decrease in phenylalanine/tyrosine/tryptophan biosynthesis (*p* = 0.032, power = 0.314) and lysine biosynthesis (*p* = 0.040, power = 0.160) at the sites with periodontal pockets were indicated (Figure 5). We also found that alanine/aspartic acid/glutamic acid metabolism decreased (*p* = 0.004, power = 0.439) and high abundances of the genes belonging to replication and repair, nucleotide excision repair (*p* = 0.024, power = 0.177), and mismatch repair (*p* = 0.005, power = 0.378) at the sites with periodontal pockets.

## 3. Discussion

In the present study, *Fusobacterium*, *Porphyromonas*, *Prevotella*, *Actinomyces*, *Treponema*, *Corynebacterium*, and *Neisseria* were predominant at the sites with periodontal pockets (Appendix A). Among 11 increased genera, *Bacteroides, Porphyromonas, Tannerella, Treponema*, and *Filifactor* have been reported as the core bacterial genera in the sites with periodontal pockets [13]. Moreover, the association between *Fretibacterium* and periodontitis has also been reported [21]. The observed shift of genera in the subgingival microbiome at the sites with periodontal pockets was consistent with the previous results. At the healthy sites of the patients with periodontitis, *Dialister, Filifactor, Fretibacterium*, and *Parvimonas* were highly abundant compared with the healthy sites of the periodontally healthy subjects (Appendix A). Increases in these species have been reported at the sites with periodontal pockets [13,22,23]. A previous report indicated that the disease-associated bacteria were detected not only in deep periodontal pockets but also in shallow pockets in individuals with periodontitis [14]. The results suggested that the shift of the microbiome precedes the appearance of symptoms. 

Among the genera with increased abundances at the sites with periodontal pockets, a significant increase in the relative abundance of *Bacteroides* and decrease in that of *Mycobacterium* at the sites with periodontal pockets were noted compared with their abundances in the healthy sites. Previous studies have also reported a significantly high abundance of *Bacteroides* in the sites with periodontal pockets [13,14,24]. *Bacteroides* detected from the sites with periodontal pockets in these studies and the present study have yet to be isolated and characterized. Because the species in *Bacteroides* are implicated in dysbiosis, the isolation and detailed characterization of these species is required*. Mycobacterium tuberculosis*, a causative bacterium of tuberculosis, is a well-known representative of this genus. However, a recent study revealed that a member of the genus *Mycobacterium,* which does not cause tuberculosis, is present in the oral cavity of healthy humans [25]. Bacteria that belong to the genus *Mycobacterium* are generally aerobic. Thus, it is conceivable that their population was reduced in the subgingival milieu, which is mostly anerobic. This genus was not predominant in the microbiome at the healthy sites, although significantly decreased in the sites with periodontal pockets. The difference in the abundance of *Mycobacterium* between the healthy sites and the sites with periodontal pockets has not reported previously. Further analysis of the prevalence of this genus concerning geographical or ethnical difference is required to clarify the association with the dysbiosis.

In the analysis of functional genes, the abundance of genes related to carbohydrate metabolism was decreased in the sites with periodontal pockets. Species in most of the increased genera listed above were asaccharolytic or weak saccharolytic [26,27,28]. These bacteria gain their nutrients from the broken cells and gingival crevicular fluid (GCF). A decrease in the abundance of functional genes for glycolysis_gluconeogenesis and galactose metabolism at the sites with periodontal pockets has been reported [29]. This decrease suggested that the major energy source of the microbiome in the sites with periodontal pockets shifts from carbohydrate to amino acid or intermediate metabolites from microorganisms in the consortia. 

Glycan biosynthesis and metabolism were increased at the sites with periodontal pockets. A previous study reported that the clinical isolates of *Prevotella intermedia* strain possessing a strong ability to produce exopolysaccharide (EPS) showed enhanced abscess-forming activity compared with low EPS producing strains [30]. In *P. gingivalis,* inactivation of *sinR*, which negatively regulates lipopolysaccharide and capsular polysaccharide synthesis*,* increased the biofilm formation [31]. The capsular polysaccharide of *P. gingivalis* plays an important role in evasion from the host defense [32]. These reports indicate that polysaccharide synthesis is important for this microorganism in the biofilm formation and evasion from host defense. *T. forsythia* is highly associated with the severe form of periodontitis. It has a specific glycan core on the cell surface. This glycan core plays an important role in the suppression of human Th17 cells [33]. The abundance of these three species was significantly increased at the sites with periodontal pockets. The observed increase in the glycan biosynthesis and metabolism suggested the importance of polysaccharides in periodontopathic biofilms.

In the amino acid metabolism category, abundance of genes involved in phenylalanine/tyrosine/tryptophan biosynthesis was decreased. In the subgingival pocket, microorganisms were bathed in GCF, which included a mixture of proteins such as albumin and globulin. The protein concentration of GCF at the sites with periodontal pockets was similar to that in the serum [34]. The predominant species, such as *P. gingivalis* and *T. denticola,* possess arginine- and lysin-specific proteases and prolyl-phenylalanine-specific protease, respectively [35,36]. In the presence of these proteolytic enzymes, lysin, phenylalanine, tyrosine and tryptophan may be accessible without synthesis. Although it was not statistically significant, a reduction in phenylalanine/tyrosine/tryptophan biosynthesis was also previously reported [23]. In addition, the amount of phenylalanine/tyrosine/tryptophan reportedly increased in the saliva of the patients with periodontitis [37]. The reduction in the genes belonging to the biosynthesis of lysine, phenylalanine, tyrosine, and tryptophan may depend on the supply of a large quantity of proteins from the host tissue. In the previous metagenomic analysis, phenylalanine-specific permease was significantly decreased at the sites with periodontal pockets, although phenylalanine/tyrosine/tryptophan biosynthesis did not change [38]. It suggested some difference in the metabolism of the phenylalanine at the sites with periodontal pockets. Interestingly, a decrease in phenylalanine, tyrosine and tryptophan biosynthesis was reported in the gut microbiome of mice, after they were administrated *P. gingivalis* [39]. The decrease in the abundance of functional genes associated with phenylalanine/tyrosine/tryptophan biosynthesis might be a characteristic of the *P. gingivalis-*induced microbiome shift. A significant reduction in phenylalanine in saliva has been observed by debridement of supragingival plaque from individuals with periodontitis [40], suggestive of the phenylalanine supply from the supragingival plaque. Further analysis of the proteome of the subgingival microbiome in periodontitis is required.

Abundance of genes associated with alanine/aspartic acid/glutamic acid metabolism also decreased in the sites with periodontal pockets. In the microbiome, complicated synergy in metabolism was organized among the organisms in the microbiome. Previous reports indicated the abundance of *P. gingivalis, T. denticola* and *T. forsythia* increase in periodontitis [7], and synergy among the species was reported [41,42]. The genes involved in the butyric acid metabolism were complemented in the three species, although, individually, they did not have the genes for the complete pathway [17]. A synergistic effect on biofilm formation between *P. gingivalis* and *Fusobacterium nucleatum* has been reported [43]. The decrease in alanine/aspartic acid/glutamic acid metabolism may be associated with the synergy among multiple species of bacteria. The evidence is limited for the metabolism of the entire microbiome in the oral cavity. Clarification of the change in the abundance of the functional genes requires further transcriptome and proteome analyses of the total microbiome.

High abundances of the genes belonging to replication and repair, nucleotide excision repair, and mismatch repair at the sites with periodontal pockets were observed. In the subgingival microbiome, bacteria are exposed to many types of stress, including oxygen stress [44]. Oxygen stress is induced by oxygen and H_2_O_2_ produced from microorganisms and neutrophils in the subgingival milieu [45,46]. Without enzyme neutralizing the H_2_O_2_, such as catalase, reactive oxygen species damage the bacterial DNA. In the present study, most genera with an increased relative abundance in the sites with periodontal pockets were anaerobes, and they do not contain catalase. In *P. gingivalis,* the gene involved in the repair of oxidative stress-induced DNA mismatching was reported [47]. It is possible that the genes involved in response to oxygen stress may partially reflect the increase in the abundance of genes associated with the nucleotide excision repair.

The increase in nucleotide excision repair suggested the involvement of the response to host defense response to functional genes of subgingival bacteria. Crosstalk between periodontopathic bacteria and immune response has been investigated. Macrophage isolated from the patients with periodontitis showed the less ability to polarize to M1 or M2 phenotype compared with that from the healthy subjects [48]. Polymorphisms of IL-10 have been associated with those of tumor necrosis factor α IL-1α, IL-1β and IL-1RA in periodontitis [49]. In patients with periodontitis, the response of neutrophil and monocytes against *P. gingivalis*-derived lipopolysaccharide was exaggerated and the immune alterations associated with periodontitis were lost after treatment of periodontitis [50]. The attenuation of the immune response was suggested to play an important role in dysbiosis [11]. Recent analysis using mice periodontitis model indicated the decrease in the amount of dental plaque and change of the microflora composition by the inactivation of IL-17 using a mice periodontitis model [51]. Investigation of the immune response with the functional analysis requires the further analysis of the meaning of the shift of the functional genes.

In the present study, the abundance of genes associated with the virulence of bacteria did not increase. Previous studies have indicated an increase in the abundance of genes associated with chemotaxis and motility, and two component system in the sites with periodontal pockets [23,52]. Here, part of the healthy site samples were collected from the patients diagnosed with periodontitis. Previous studies indicated that the relative abundances of functional genes and taxa are different between the healthy sites in healthy subjects and the gingival crevice or deep periodontal pocket in patients with periodontitis, but are not different between the gingival crevice and deep periodontal pocket in patients with periodontitis. In the microbiome composition found in the present study, abundances of genera, such as *Bacteroides, Tannerella, Fretibacterium*, and *Filifactor*, frequently increased in periodontitis. It is possible that a subliminal increase in periodontopathic bacterial abundance attenuated the detection of virulence-associated genes.

Nonetheless, there are some limitations to this study. The metagenomic analysis is limited owing to its biases [53]: bias from the extraction of the gene sequence from the sample and methodology used for bioinformatics analysis of taxonomy and sequence depth (the method cannot detect < 10^5^ per gram of sample). In this study, we aimed to obtain basic information on the potential difference between healthy sites and sites with periodontal pockets, and sample size calculation was not performed a priori. Therefore, it is possible that we could not find some of the features to be significant because of a type II error. Although this method is advantageous in investigating a large number of microbiome compositions in a short time, the determination of the genera and functional gene abundances was performed without the isolation of bacteria. To complement the metagenomic data, a culture-based approach, such as culturomics, is required [12]. Despite these limitations, our study provided important implications in the shift of functional gene abundances occurring during the progression of periodontal disease. 

In conclusion, the taxonomic abundances were higher in sites with periodontal pockets than those in healthy sites. In the functional gene categories, carbohydrate metabolism, glycan biosynthesis and metabolism, amino acid metabolism, replication and repair showed significantly different abundances between healthy sites and sites with periodontal pockets. Among them, a decrease in phenylalanine/tyrosine/tryptophan biosynthesis was suggested as a candidate marker of dysbiotic shift in periodontitis. These differences might be useful for clarifying the mechanisms of the microbiome shift occurring in periodontitis. Future metagenomic research in larger Japanese cohorts combined with transcriptome and metabolome analyses should provide us with more information about the changes in taxonomic and functional gene category abundances caused by periodontitis.

## 4. Materials and Methods

### 4.1. Sample Collection

Systemically healthy individuals (aged 20 to 80 years) were recruited from the Tokyo Dental College hospitals. Subjects who had a history of antimicrobial agent use for the past 3 months, smoking, or serious underlying disease were excluded. We followed the description in the literatures for the diagnosis of periodontitis [54,55]. Periodontally healthy subjects showed no probing depth > 3 mm, and no attachment loss or detectable inflammation. Subjects with periodontitis had at least six sites with a probing depth >5 mm and two sites exhibiting bleeding upon probing. Subgingival plaque samples were collected from volunteers with healthy periodontium and patients with periodontitis. From patients with periodontitis, samples were obtained from healthy and diseased sites. After removal of supragingival plaque, the site was isolated with cotton rolls, gently air-dried, and subgingival plaque were collected with scaler. The obtained samples were suspended in PowerBead tubes of Power Soil DNA Isolation kit (MoBio, Carlsbad, CA, USA) and stored at −20 °C until DNA isolation. All participants provided written informed consent. Ethical approval for this study was obtained from the Tokyo Dental College Ethics Committee (No. 622).

### 4.2. DNA Extraction, Library Preparation and Sequencing

DNA was extracted from each sample using a Power Soil DNA Isolation kit (MoBio) according to the manufacturer’s protocol. The extracted DNA samples were quantified using a Quantus Fluorometer (Promega, Madison, WI, USA). One nanogram of input DNA was enzymatically fragmented and tagged with sequencing adapters using the Nextera XT DNA Library Preparation Kit (Illumina, San Diego, CA, USA) and then normalized.

Sequencing was conducted using the Illumina MiSeq with the MiSeq Reagent Kit v2 (300 cycles).

### 4.3. Sequencing Data Analysis

The obtained reads were subjected to quality control using PRINSEQ v0.20.4 [56] to obtain a quality score of more than 20 in all parts of the reads. The sequencing tags were trimmed using cutadapt v1.14 [57]. The reads derived from the human genome were excluded by mapping them to the human genome (GRCh38) using bowtie2 v2.3 [58]. Following the literature [59], samples retaining more than 200,000 reads were subjected to further analysis.

For taxonomic analysis at the genus level, the 16S ribosomal RNA database distributed by NCBI was downloaded in February 2019 and used. We aligned the reads to the database using BLASTN [60] in default settings. The relative taxonomic abundance was calculated using MEGAN6 [61].

For functional gene category analysis, the KEGG gene database [62] distributed by KEGG was downloaded in December 2019 and used. We aligned reads to the database using GHOSTZ-GPU [63] on TSUBAME3.0 in default settings. The relative abundance of functional gene categories was calculated using HUMAnN [64].

The obtained relative abundance data were subjected to a two-sided Mann–Whitney *U* test with Bonferroni correction. We set the significance level at *p* < 0.05 and calculations were conducted using the stats module implemented by SciPy v1.5. For convenience, the *p*-values indicated in the text were *p-*values from a single test multiplied by the number of tests. The statistical powers were calculated using G*power software [65].

## Figures and Tables

**Figure 1 ijms-22-05298-f001:**
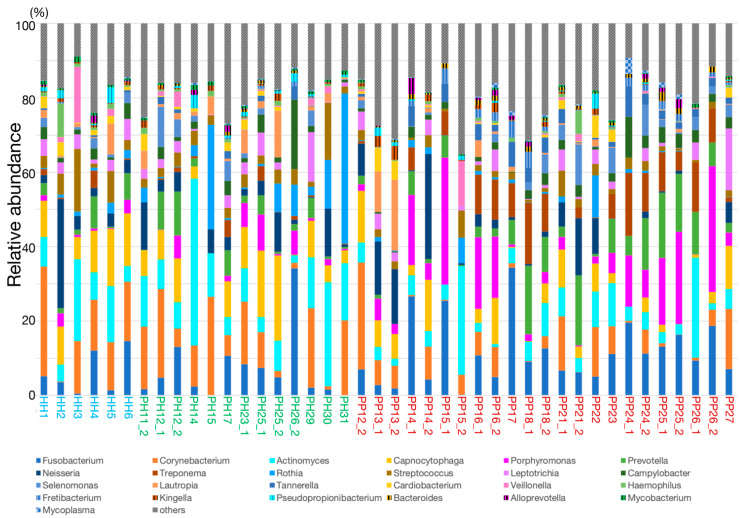
The genus level taxonomic composition in the top 25 genera, representing the mean of all samples.

**Figure 2 ijms-22-05298-f002:**
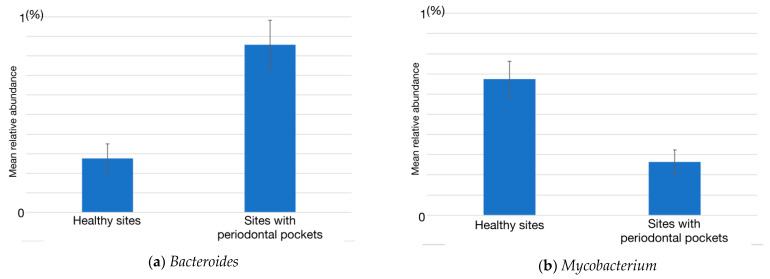
The relative abundances of the genera *Bacteroides* (**a**) and *Mycbacterium* (**b**). The relative abundance of *Bacteroides* and *Mycobacterium* were significantly different between the healthy sites and sites with periodontal pockets (*p* < 0.05). Data are shown as the mean ± standard error of the mean.

**Figure 3 ijms-22-05298-f003:**
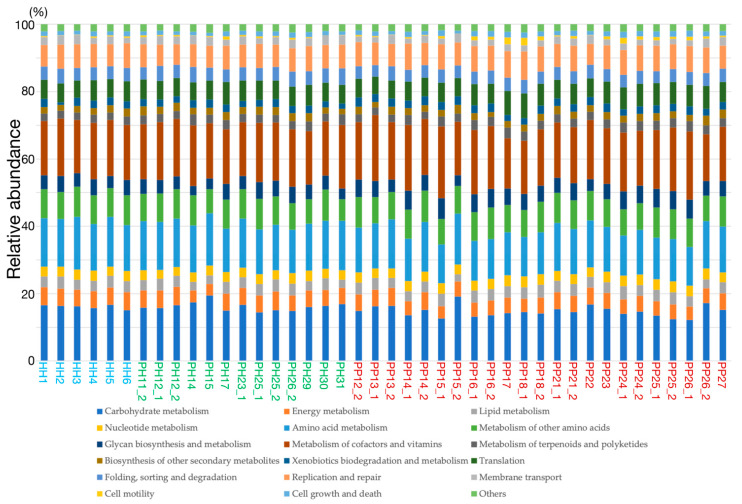
The relative abundances of functional gene categories.

**Figure 4 ijms-22-05298-f004:**
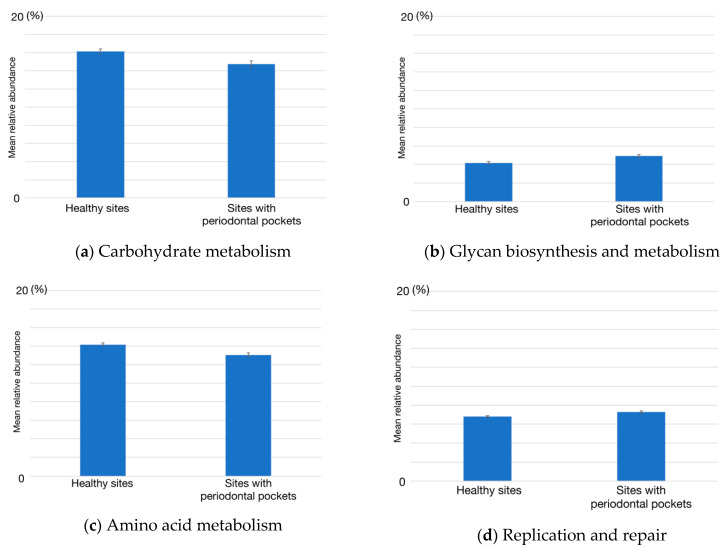
Relative abundances of four functional gene categories in the healthy sites and sites with periodontal pockets. (**a**) Carbohydrate metabolism, (**b**) glycan biosynthesis and metabolism, (**c**) amino acid metabolism, and (**d**) replication and repair. The four functional gene categories were significantly different between healthy and periodontitis sites (*p* < 0.05). Data are shown as the mean ± standard error of the mean.

**Figure 5 ijms-22-05298-f005:**
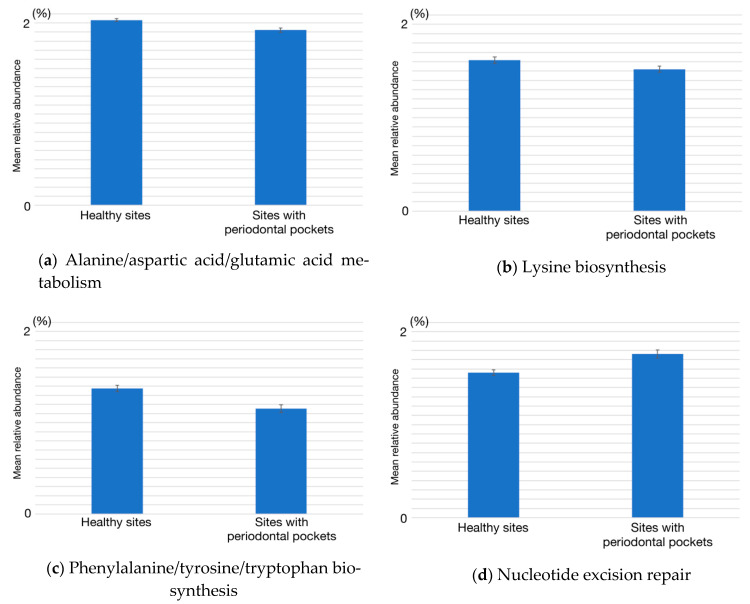
The relative abundances of five lower functional gene categories in the healthy sites and sites with periodontal pockets. (**a**) Alanine/aspartic acid/glutamic acid metabolism, (**b**) lysine biosynthesis, (**c**) phenylalanine/tyrosine/tryptophan biosynthesis, (**d**) nucleotide excision repair, and (**e**) mismatch repair. The five functional gene categories were significantly different between healthy and periodontitis sites (*p* < 0.05). Data are shown as the mean ± standard error of the mean.

**Table 1 ijms-22-05298-t001:** Sequence result overview.

Sample Name	Indiv. No.	Status of Individual	Sampling Site and Status	Age	Sex ^1^	Raw Reads	Reads for Metagenomic Analysis
HH1	1	Healthy	46, Healthy	29	M	5,775,202	1,402,887
HH2	2	Healthy	46, Healthy	43	M	3,831,266	204,308
HH3	3	Healthy	46, Healthy	37	M	5,422,140	995,685
HH4	4	Healthy	46, Healthy	41	F	4,866,418	1,100,826
HH5	5	Healthy	46, Healthy	31	F	4,247,496	1,089,164
HH6	6	Healthy	46, Healthy	27	M	4,265,064	663,850
PH11_2	11	Periodontitis	43, Healthy	48	M	2,624,566	548,873
PH12_1	12	Periodontitis	23, Healthy	69	M	1,205,826	695,215
PH12_2	12	Periodontitis	25, Healthy	69	M	2,697,604	1,750,204
PH14	14	Periodontitis	43, Healthy	48	F	8,165,844	239,717
PH15	15	Periodontitis	12, Healthy	79	M	7,358,454	366,238
PH17	17	Periodontitis	24, Healthy	56	F	2,125,774	278,004
PH23_1	23	Periodontitis	23, Healthy	42	M	2,340,430	463,933
PH25_1	25	Periodontitis	41, Healthy	43	M	5,952,692	1,716,829
PH25_2	25	Periodontitis	41, Healthy	43	M	4,593,914	1,556,673
PH26_2	26	Periodontitis	31, Healthy	60	M	4,678,568	225,769
PH29	29	Periodontitis	27, Healthy	58	F	5,253,782	516,845
PH30	30	Periodontitis	13, Healthy	76	M	6,012,650	390,445
PH31	31	Periodontitis	23, Healthy	46	F	4,890,982	213,660
PP12_2	12	Periodontitis	25, Periodontal pocket	69	M	2,398,604	1,894,262
PP13_1	13	Periodontitis	26, Periodontal pocket	47	F	327,016	245,066
PP13_2	13	Periodontitis	25, Periodontal pocket	47	F	309,300	306,024
PP14_1	14	Periodontitis	16, Periodontal pocket	48	F	7,440,288	535,904
PP14_2	14	Periodontitis	27, Periodontal pocket	48	F	7,213,688	630,156
PP15_1	15	Periodontitis	11, Periodontal pocket	79	M	7,346,724	414,834
PP15_2	15	Periodontitis	47, Periodontal pocket	79	M	6,530,194	350,097
PP16_1	16	Periodontitis	24, Periodontal pocket	55	F	1,904,568	452,740
PP16_2	16	Periodontitis	26, Periodontal pocket	55	F	3,093,452	226,275
PP17	17	Periodontitis	47, Periodontal pocket	56	F	3,646,976	297,839
PP18_1	18	Periodontitis	16, Periodontal pocket	36	F	1,604,552	439,199
PP18_2	18	Periodontitis	25, Periodontal pocket	36	F	3,920,256	912,649
PP21_1	21	Periodontitis	27, Periodontal pocket	50	M	1,099,140	998,367
PP21_2	21	Periodontitis	13, Periodontal pocket	50	M	2,426,890	987,197
PP22	22	Periodontitis	37, Periodontal pocket	48	F	7,794,000	1,322,677
PP23	23	Periodontitis	35, Periodontal pocket	42	M	8,557,728	1,484,973
PP24_1	24	Periodontitis	13, Periodontal pocket	59	F	5,485,608	262,343
PP24_2	24	Periodontitis	16, Periodontal pocket	59	F	8,658,802	768,780
PP25_1	25	Periodontitis	27, Periodontal pocket	43	M	9,030,826	2,508,972
PP25_2	25	Periodontitis	47, Periodontal pocket	43	M	6,575,378	817,066
PP26_1	26	Periodontitis	18, Periodontal pocket	60	M	1,427,528	1,008,404
PP26_2	26	Periodontitis	37, Periodontal pocket	60	M	11,622,702	440,551
PP27	27	Periodontitis	24, Periodontal pocket	56	F	8,580,870	3,233,325

^1^ M, male; F, female.

## Data Availability

The read data analyzed in this study were deposited in the DNA Data Bank of Japan under the accession numbers DRR285642-DRR285683.

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
