# Peer review of "Taxonomic and Gene Category Analyses of Subgingival Plaques from a Group of Japanese Individuals with and without Periodontitis"

_ijms, 2021, doi:10.3390/ijms22105298_

Round 1
Reviewer 1 Report
The authors aimed to present a metagenomics analysis of subgingival plaque samples from a group of Japanese individuals with or without periodontitis, aiming to clarify the taxonomic and functional genomic abundance in periodontitis.
The structure of the manuscript appears adequate and correctly divided in sub-paragraphs. The methodology is well described with enough experimental data and results to support the work. Please check typos thorough the text.
Results and Discussion section: Will be very useful for the readers to stress better the crosslink with oral microbial dysbiosis in the progression of the disease/systemic diseases (please see and discuss: PMID: 32397555; PMID: 32560235; PMID: 24843315; PMID: 20646367).
Conclusion Section: This paragraph required a general revision to eliminate redundant sentences and to add some "take-home message".
Author Response
We thank the editor and reviewers for the helpful comments. We revised our manuscript according to their suggestions. Our point-by-point responses are listed below.
The authors aimed to present a metagenomics analysis of subgingival plaque samples from a group of Japanese individuals with or without periodontitis, aiming to clarify the taxonomic and functional genomic abundance in periodontitis.
The structure of the manuscript appears adequate and correctly divided in sub-paragraphs. The methodology is well described with enough experimental data and results to support the work. Please check typos thorough the text.
(Response)
Thank you for the positive comments and advice. We revised the typos throughout the text again.
Results and Discussion section: Will be very useful for the readers to stress better the crosslink with oral microbial dysbiosis in the progression of the disease/systemic diseases (please see and discuss: PMID: 32397555; PMID: 32560235; PMID: 24843315; PMID: 20646367).
(Response)
Thank you for your suggestion we added the references PMID: 32397555; PMID: 32560235 and added such contents in the Discussion section. Please see the revised manuscript. (Line 270-284)
Conclusion Section: This paragraph required a general revision to eliminate redundant sentences and to add some "take-home message".
(Response)
Thank you for your advice. We revised the conclusion, accordingly. (Line 311-321)
Reviewer 2 Report
This study focuses on an aspect of periodontitis on which further research is still needed. The work is interesting and in general the manuscript is well written although some points of the methodology can be improved.
In the version I received the results come after the materials and methods. Formatting should be revised as there are different fonts throughout the manuscript The abstract could be structured by sections. M&M How was the sample size calculated? What are the inclusion and exclusion criteria? Did the study include smokers? The criteria for defining disease/health could be described in the text. One of the bibliographic references used does not seem to be the most suitable for this purpose. How were the samples collected and stored? Authors should use the term locations with pockets instead of locations with periodontitis. The periodontitis diagnosis concerns to the patient.The nomenclature used also creates some confusion throughout the text. Results and discussion The limitations of this study could be better discussed. The comparison of the results obtained with other studies should be further explored.Author Response
We thank the editor and reviewers for the helpful comments. We revised our manuscript according to their suggestions. Our point-by-point responses are listed below.
This study focuses on an aspect of periodontitis on which further research is still needed. The work is interesting and in general the manuscript is well written although some points of the methodology can be improved.
In the version I received the results come after the materials and methods. Formatting should be revised as there are different fonts throughout the manuscript
(Response)
Thank you for your advice. I revised the formatting and the fonts used.
The abstract could be structured by sections.
(Response)
“Instruction for authors” of International Journal of Molecular Sciences does not require the structured abstract. We followed the format.
M&M How was the sample size calculated?
(Response)
Because of the nature of this study to find some basic information on the difference between healthy sites and sites with periodontal pocket, we did not calculate sample size for this study a priori. In light of the reviewer’s comment, we added the a posteriori statistical power for each significant feature. As a result, these power values were low, suggesting that we overlooked some features by type II error. Therefore, we mentioned this as a limitation to the present study in the Discussion section. (Line 301-305)
What are the inclusion and exclusion criteria? Did the study include smokers?
The criteria for defining disease/health could be described in the text. One of the bibliographic references used does not seem to be the most suitable for this purpose.
(Response)
The study excluded smokers. We clarified inclusion and exclusion criteria in “4.1. sample collection” and changed the reference. (Line 326-331)
.
How were the samples collected and stored?
(Response)
We added such information in the revised manuscript. (Line 333-337)
Authors should use the term locations with pockets instead of locations with periodontitis. ?
(Response)
Although the diagnosis of periodontitis is made not only at patient level but also at site (tooth) level, we made the necessary changes, in order to avoid the confusion. We changed them to “sites with periodontal pockets”, according to the advice from the reviewer We also change the descriptions in the Table.
The periodontitis diagnosis concerns to the patient. The nomenclature used also creates some confusion throughout the text. Results and discussion.
(Response)
Thank you for this advice. Although the diagnosis of periodontitis is made not only at patient level but also at site (tooth) level, we made the necessary changes, in order to avoid the confusion. In the Results and Discussion, we decided to use the expressions; “healthy sites”, “sites with periodontal pockets”, “healthy subjects”, and “patients with periodontitis”.
The limitations of this study could be better discussed.
(Response)
We revised the limitation sentences. (Line 298-310)
The comparison of the results obtained with other studies should be further explored.
(Response)
Thank you for your advice. The number of the metagenomic analysis of periodontitis is small. We added the previous metagenomic works, metabolome analysis and compared additional points such as virulence associated genes in the discussion. (Line 197-297)
Reviewer 3 Report
Dear authors,
even if the manuscript I reviewed is very interesting and deals with an emerging topic nevertheless it has some methodological inconsistencies so I'd suggest a round of major revision before going ahead in the editorial process.
Here you can find in detail all my main concerns:
- In introduction you should describe the possibility to use other analysis techniques such as culturomics citing appropriate literature (for your convenience: Martellacci L et al. A literature review of metagenomics and culturomics of the peri-implant microbiome: Current evidence and future perspectives. Materials. 2019;12(18):3010 AND Martellacci L et al. Characterizing peri-implant and sub-gingival microbiota through culturomics. First isolation of some species in the oral cavity. A pilot study. Pathogens. 2020;9(5):365);
- The same concepts should be expressed also in discussion when dealing about metagenomic limitations (e.g. metagenomics cannot discriminate between bacterial species) that could be by-passed by the use of culturomic approach implemented by quantitative PCR;
- Line 71 in introduction: "Aggressive parodontitis" is a no longer accepted term according to the new Periodontal Disease Classification, please delete;
- Please separate the two paragraphs about results and discussion as it is very hard to understand by the reader;
- English needs to be strongly revised by a native speaker as there are a lot of odd sentences and errors throughout the whole manuscript;
- In paragraph 3.1 (belonging to Materials and Methods) no data must be reported so please delete all and report in the appropriate section (results);
- Please give a power analysis for sample size calculation.
Regards
Author Response
We thank the editor and reviewers for the helpful comments. We revised our manuscript according to their suggestions. Our point-by-point responses are listed below.
- In introduction you should describe the possibility to use other analysis techniques such as culturomics citing appropriate literature (for your convenience: Martellacci L et al. A literature review of metagenomics and culturomics of the peri-implant microbiome: Current evidence and future perspectives. Materials. 2019;12(18):3010 AND Martellacci L et al. Characterizing peri-implant and sub-gingival microbiota through culturomics. First isolation of some species in the oral cavity. A pilot study. Pathogens. 2020;9(5):365);
- The same concepts should be expressed also in discussion when dealing about metagenomic limitations (e.g. metagenomics cannot discriminate between bacterial species) that could be by-passed by the use of culturomic approach implemented by quantitative PCR.
(Response)
In light of the reviewer’s advice, we added the possibility to use other analyses in the Introduction section. We also revised the description regarding the study limitations, in the Discussion section (please see the revision in the main text). (Line 57-67, 310)
3. Line 71 in introduction: "Aggressive parodontitis" is a no longer accepted term according to the new Periodontal Disease Classification, please delete;
(Response)
We removed “aggressive”.
4. Please separate the two paragraphs about results and discussion as it is very hard to understand by the reader.
(Response)
Thank you for your advice. We separated the results and discussion sections.
5. English needs to be strongly revised by a native speaker as there are a lot of odd sentences and errors throughout the whole manuscript.
(Response)
Although the use of English was checked by the professional English editing service, the revised manuscript was re-checked by the service.
- In paragraph 3.1 (belonging to Materials and Methods) no data must be reported so please delete all and report in the appropriate section (results).
(Response)
Thank you for this comment. We removed the data from the section and moved them to the Results section.
7. Please give a power analysis for sample size calculation.
(Response)
Because of the nature of this study to find some features distinguish healthy sites and sites with periodontal pocket, we did not calculate sample size for this study a priori. Then, we added the a posteriori statistical power for each significant feature. As a result, some values for statistical power were found to be low, indicating that we could have overlooked some features by type II error. We mentioned this as another limitation of the study in the Discussion section. (Line 301-305)
Round 2
Reviewer 3 Report
Dear authors,
thanks for having provided all the suggested changes.
I suggest the article to go ahead in its editorial process.
Regards